# Primary Clear Cell Adenocarcinoma of the Uterine Cervix in a 14-Year-Old Virgin Girl: Case Report

**DOI:** 10.3390/ijerph192416652

**Published:** 2022-12-11

**Authors:** Iuliana Elena Bujor, Ludmila Lozneanu, Alexandra Ursache, Alexandra Cristofor, Ana-Maria Scurtu, Petru Plamadeala, Roxana Gireada, Cristina Elena Mandici, Marcel Alexandru Găină, Daniela Roxana Matasariu

**Affiliations:** 1Department of Obstetrics and Gynecology, University of Medicine and Pharmacy ‘Grigore T. Popa’, 700115 Iasi, Romania; 2Department of Morpho-Functional Sciences I—Histology, Pathology, “Grigore T. Popa” University of Medicine and Pharmacy, 700115 Iasi, Romania; 3Department of Obstetrics and Gynecology, Cuza Vodă Hospital, 700038 Iasi, Romania; 4Department of Pediatric Surgery, Children’s Clinical Hospital “St. Maria”, 700309 Iasi, Romania; 5Department of Pathology, Children’s Clinical Hospital “St. Maria”, 700309 Iasi, Romania; 6Psychiatry, Department of Medicine III, “Grigore T. Popa” University of Medicine and Pharmacy, 700115 Iasi, Romania

**Keywords:** clear cell adenocarcinoma, virgin girl, non-HPV, diethylstilbestrol, radiotherapy, chemotherapy, intracavitary brachytherapy, immunohistochemical

## Abstract

Cervical cancer is rare in adolescent and pediatric populations, with adenocarcinoma being the most commonly reported. Clear cell adenocarcinoma of the uterine cervix accounts for only 4% of all adenocarcinoma cases, and about two-thirds are associated with intrauterine diethylstilbestrol (DES) exposure. We report the case of a 14-year-old virgin girl who presented with a 1-month-long history of abnormal vaginal bleeding and lower abdominal pain. Transabdominal pelvic ultrasound examination revealed the presence of an irregular, homogeneous cervical mass that was 7 cm in size. Therefore, a magnetic resonance imaging (MRI) scan was performed to establish the origin of the tumor and its relationship to adjacent pelvic organs. Furthermore, a vaginoscopy was performed to identify the tumor, and a cervical biopsy was performed. Immunohistochemical and anatomopathological studies resulted in the diagnosis of non-HPV(Human Papilloma Virus)-related clear cell adenocarcinoma of the cervix. Following the oncological examination, she was admitted for radiotherapy. The patient had no maternal history of DES exposure in utero. Even though the number of cases in the literature is low, most of the virgin girls diagnosed with clear cell adenocarcinoma of the cervix have a fatal prognosis because of the delay in making a correct diagnosis.

## 1. Introduction

Worldwide, cervical cancer is the fourth most frequent cancer in women and the most common gynecologic cancer [1].

Histologically, the two most frequent subtypes of cervical cancer are squamous cell carcinoma (75%) and adenocarcinoma (20–25%). Clear cell carcinoma of the uterine cervix is an exceedingly rare and highly invasive variant of adenocarcinoma and accounts for only 4% of all adenocarcinomas of the uterine cervix. This type of adenocarcinoma can be also found in other sites of the female genital tract, such as in the vagina, ovaries, and endometrium. Its etiology and pathogenesis remain unclear, but many reports have associated this cancer type with prior in utero diethylstilbestrol (DES) exposure. Nevertheless, cases of clear cell adenocarcinoma have been reported in both adult women and in children without a history of DES exposure. Cervical cancer is uncommon in childhood and adolescence, and adenocarcinoma of the cervix and vagina is rare. There are less than 50 cases reported in the literature [2,3]. Compared to adults who suffer from adenocarcinoma, there is a high incidence of stage III and stage IV disease in children and adolescents [4].

We report the pathology findings and clinical features in a young patient with stage IIIC1 sporadic, non-HPV (Human Papilloma Virus)-related primary cervical clear cell adenocarcinoma according to the revised 2018 FIGO (International Federation of Gynecology and Obstetrics) staging criteria for cervical cancer [5].

Informed consent for the publication of this case report was offered by the mother of our underage adolescent patient, and this case report was approved by the Ethics Committee of “Cuza Voda” University Hospital (14189/25.10.2022).

## 2. Case Presentation

A 14-year-old girl presented to our hospital because of continuous vaginal bleeding for about 1 month. The patient had her first period at the age of 12, with a regular menstrual cycle (30–32 days) that lasted for 7 days without any other medical conditions or subjective complaints.

She was evaluated two times prior to hospital presentation for the same complaints and received tranexamic acid in an attempt to improve the symptoms, but with no visible results.

This time, the decision was to perform an ultrasound evaluation of the patient, and because the patient was not yet sexually active, a transabdominal ultrasound was performed. It revealed a 7 cm irregular homogeneous hyperechoic mass in the pelvis. Digital rectal examination confirmed the presence of the tumor and also identified partial parametrial involvement.

### 2.1. Magnetic Resonance Imaging (MRI) Findings

A pelvic MRI was recommended to our patient, as it would be able to provide an accurate anatomic localization of the mass and its relationship with the surrounding organs. It revealed an expansive solid cervical mass that developed between the internal and external cervical orifice, occupying the left vaginal recess, with imprecise boundaries being observed at this level. Axial and sagittal T2-weighted images showed a 53 × 78 × 46 mm mass that was hyperintense compared to normal myometrium, and axial T1-weighted images showed an isointense mass compared to the myometrium that was occupying the uterine cervix and upper vagina. The ovaries and the uterine body were normal in size, and a right external iliac adenopathy was also detected (Figure 1).

### 2.2. Computer Tomography (CT) Findings

Subsequently, a thoracoabdominal CT scan was performed that revealed the absence of secondary lesions in the lungs, liver, or bone in the scanned area.

### 2.3. Supplementary Investigations

Two days after performing the MRI, the patient was admitted to the hospital with extreme fatigue, pain in the lower abdomen, and heavy genital bleeding. On admission, she was in a poor general condition, with pale skin and tachycardia. The blood tests showed severe microcytic hypochromic anemia (hemoglobin concentration—4.9 g/dL), thrombocytosis (530,000 mm^3^), and hyposideremia. Tumor marker levels (lactate dehydrogenase, α fetoprotein, human chorionic gonadotropin, and carbohydrate antigen CA-125) were within normal ranges. During her hospital stay, the patient received repeated erythrocyte mass transfusions.

Under general anesthesia, a urinary endoscopy was performed, through which a supratrigonal bladder wall displaced by an extravesical tumor was detected, and a follow-up vaginoscopy showed that the vagina was occupied by clots, blood, and a cauliflower-like mass with a fragile and necrotic surface filing the upper vagina. The cervix could not be visualized because of the clots and the bulky tumoral mass. A punch biopsy of the cauliflower-like mass was performed.

### 2.4. Histopathology Findings

The microscopic analysis of the cervical mass biopsy showed the presence of the following morphological aspects: mixed architecture with solid and tubulocystic patterned areas; mainly composed of nests of cells with clear or eosinophilic cytoplasm (intracytoplasmic glycogen); and prominent hyperchromatic nuclei, some with round nucleoli and high mitotic activity (16/10 HPF) (Figure 2, Figure 3 and Figure 4). Additionally, we noted abundant inflammatory infiltrate that was mainly made up of the lymphocytes and plasmacytes present in the hyalinized stroma, and this was determined to be associated with the area of necrosis. Periodic acid–Schiff staining was strongly positive in the cytoplasm, consistent with glycogen (Figure 5). Immunohistochemical staining showed a diffuse positive reaction for CK AE1/AE3, PAX8, and AMACR; rare cells showed CEA-positive expression; and there was a negative reaction for p63, CD30, OCT4, ER, SALL4, synaptophysin, CD56, and p16, which were focally positive in tumor cells (Figure 6, Figure 7, Figure 8, Figure 9, Figure 10 and Figure 11). The histological diagnosis was HPV-negative clear cell adenocarcinoma of the uterine cervix.

According to the FIGO classification, a diagnosis of stage IIIC1 cervical cancer was established.

### 2.5. Case Management

The patient had no history of in utero DES exposure and did not receive any hormone treatment for irregular genital bleeding. She was a virgin and had no family history of cancer.

The oncologists decided to initiate concurrent chemotherapy and radiotherapy.

A genital examination performed prior to therapy described a friable tumor mass that bled spontaneously or after minor trauma occupying the entire cervix and the upper third of the vagina with partial invasion of the parametrium.

The patient underwent external radiotherapy using the VMAT technique (Clinac Varian IX linear accelerator 6 MV photon): total dose of 45 Gy/25 fr/32 days on the cervix, uterine body, the upper vagina, parametria, and lymph node chains (common, internal and external iliac lymph nodes; obturator, superior presacral and periaortic lymph nodes). For a simultaneous integrated boost, a total dose of 55 Gy/25 fr was administered on the pelvic lymphadenopathy identified on the CT scan, and a total dose of 57.5 Gy/25 fr was administered on the right external iliac adenopathy. Additionally, five weekly chemotherapy cycles with cisplatin 40 mg/m^2^ were initiated.

During the radiotherapy and chemotherapy the patient developed a grade I trombocytopenia, a grade III anemia (hemoglobin concentration 6.6 g/dL), and a grade III lymphopenia that required multiple blood transfusions for correction.

The genital examination performed after therapy described a reduction in tumor size in the exocervix compared to the initial examination, with only 1 cm extension in the right lateral vaginal sac, with no clinical signs of left parametrial invasion, and with the invasion of the right parameter.

An MRI scan performed immediately the next day after finishing the radiotherapy and chemotherapy sessions showed a significant favorable response, with the persistence of a necrotic and ulcerative cervical area and a possible residual lesion on the right side of the cervix (Figure 12).

Due to the persistent residual tumor after radiotherapy and chemotherapy (Figure 13), the patient was referred for intracavitary brachytherapy, achieving favorable evolution and almost complete response (Figure 14). She underwent three sessions of 3D intracavitary and interstitial cervico-uterine brachytherapy, after the insertion of 8 needles Ring and Tandem applicator. All three sessions took place 2 days apart in December 2021, the first one with 7.5 Gy, and the next two with 8 Gy.

The patient is on regular follow-up aimed at detecting a possible recurrence. Until the submission of the article, from April to September 2022, no sign of relapse was detected during routine CT and MRI follow-up.

## 3. Discussion

Herbst et al. study documented for the first time in 1971 the association between maternal DES therapy during pregnancy and clear cell carcinoma of the vagina and cervix in 8 cases. Another one of his studies summarized the presence of clear cell carcinoma of the vagina and cervix in 547 patients from United States, maternal treatment during pregnancy with DES or similar compounds being documented in 60% of them [6,7]. According to some studies, cervical clear cell carcinoma has two peaks of age distribution. The first one is in women aged between 17 and 37 years (mean age 26 years). In these cases, young women are often exposed to DES in utero. The second peak is observed in women aged 44 to 88 years (mean age 71 years) and mostly comprises women who have not been exposed to DES [8]. In contrast, the study conducted by Seki et al. concluded that this type of cancer occurs in all age groups, with the mean age for DES-unexposed women being 50.8 years [9]. This age distribution was confirmed by a 2008 multi-institutional review, in which the median age of the 34 patients in which cervical clear cell carcinoma was detected was 53 years, with only three of them being 30 years of age [10].

These types of tumors, without maternal history of DES exposure, are extremely rare in prepubertal age patients, the youngest reported in literature being 1-year-old [11].

We did a review of the literature by searching on PubMed, Embase and Medline for the most relevant articles/studies, including case reports, systematic reviews, and meta-analysis regarding non-HPV related cervical clear cell carcinoma in teenagers without DES exposure. We selected and included them in Table 1.

Most of these cancer types are of Müllerian origin, affecting any region of the vagina and uterine cervix, or both. The upper third of the anterior vaginal wall is a frequent location of these tumors because it is the site for the majority of vaginal adenosis lesions [34]. Diethylstilbestrol (DES) has estrogenic and anti-androgenic activity, having multiple side effects on reproductive function by interfering with the development of Mullerian ducts, with the most devastating result being cervical clear cell carcinoma [35].

The etiopathogenesis of cervical clear cell carcinoma remains hazy, but facts show that compared to squamous cell carcinoma, growth is slower, distant metastasis can be determined faster, and patients are predisposed to late recurrence. However, cervical clear cell carcinoma in pediatric populations tends to be associated with DES exposure [36], although there are cases reported in literature that come from countries where this compound was banned a long time ago or was never used, such as the United States, the Netherlands, Norway, or Japan [18]. All of these data suggest that DES may play an important but not necessarily a decisive role in the etiopathogenesis of this cancer type, with molecular genetics demonstrating that the overexpression of the p53 protein, HPV infection, bcl-2 overexpression, and microsatellite instability might be involved [16,37,38,39]. Additionally, genitourinary tract malformations are incriminated in the development of this particular disease [16]. These aspects are reflected by our case because she had no history of hormone treatment or prenatal exposure to DES, and no genitourinary tract malformations were observed.

This disease has nonspecific clinical manifestations that delay correct diagnosis. This assertion is also supported by our case. It manifests itself with irregular, prolonged vaginal bleeding that is often confused with normal hormonal axis instability that occurs during puberty. Additionally, interference in making a correct diagnosis is due to negative vaginal cytology and the impossibility of identifying the tumor on clinical examination or due to hesitation in performing an examination in the case of a negative history of sexual activity in underage adolescents. The diagnosis of cervical clear cell carcinoma must be suspected in cases of prolonged unresponsiveness to treatment for vaginal bleeding. There are a number of diseases, both benign and malignant, that might share its clinical manifestations, such as mesonephric papilloma, polyps, rhabdomyosarcoma, and endodermal sinus tumors [16].

The parameters important for determining the prognosis of this condition are tumor stage, size, growth pattern, nuclear atypia, mitotic activity, and the general appearance of the cervical mass, with tubulocystic tumors having a more favorable prognosis than solid or mixed ones. Unfavorable prognosis is suggested by larger size (>4 cm), higher stage, a high mitotic rate, a positive surgical margin, parametrial involvement, and lymphovascular spread [8,9,24,34,40]. Survival rates cannot be determined with accuracy due to the rarity of this disease, but it seems to be 90% for stage I, 71% for stage II, and 29% for stages III and IV [2]. The most frequent sites of tumor recurrence are the lungs, liver, and skeletal system, with a median for recurrence of 12 months for stages I and II of the disease [40].

Due to the negative impact of both radiation therapy and chemotherapy on the ovarian function of adolescents with stage I or II cervical clear cell carcinoma, surgery is the treatment of choice [41], while in adult women, radiation therapy and chemotherapy are preferred [15], even though the results concerning tumor response and organ preservation seem optimal [17]. Data in the literature on the association of radiation therapy and chemotherapy in children with cervical clear cell carcinoma are scarce due to hesitation in using it. Additionally, because of the extremely low incidence of this condition, the treatment of advanced stages remains unclear [17]. Our case demonstrates the success of this association in the management of such a case.

Because survivor rate is higher in early stage disease, fertility preservation can be taken into consideration in the management of such cases. Radical hysterectomy with lymph node excision remains the standard approach even in prepubertar and adolescent early stage cases. Still, surgical mass removal (radical abdominal trachelectomy) with bilateral pelvic lymph node dissection remains the best option for survival in both early stage disease young and adult patients. If the ovaries are not involved, their removal is unnecessary, with oophoropexy being recommended if postoperative radiation therapy is considered. However, this technique has no effect in protecting the ovaries from chemotherapy’s negative impact [2,17].

Even though the case reported by us was diagnosed as stage IIIC1 cervical clear cell carcinoma, radiotherapy and chemotherapy was highly successful. Both radiotherapy and cisplatin chemotherapy are known to have a negative impact on ovarian function. Fertility preservation options in such a case, with advanced stage disease and recommendation of associated radiotherapy and chemotherapy, are oocyte or ovarian tissue cryopreservation. Random start ovarian gonadotropin stimulation with 10 to 15 oocytes cryopreservation is a fertility saving recommended option. Ovarian cryopreservation technique can represent an experimental option in cases that need to immediately start their treatment. The procedure requires two unstandardized surgical interventions: partial or complete surgical removal of the ovary and secondary reimplantation. The procedure might prove being a failure due to ischemia or follicular burn-out after reimplantation [42,43].

Given the low survival rate, even in stages I and II of this disease, the frequent recurrences, and negative impact on fertility in young women, management is challenging. An optimal method for therapeutic management still remains to be established, and novel therapeutic options are being explored. In their 2001 study, Zhang et al. evaluated the possibility of using the oncolytic virus therapy Oncorine [32], and in 2020, Levinson et al. evaluated the success rate of immunotherapy with the monoclonal antibodies Nivolumab and Ipililumab [31].

In young patients, fertility preservation counseling is crucial and must be an integral part of correct management [43]. Adolescents that overcome this disease with retained ovarian function are prone to premature menopause with sexual dysfunction, increased cardiovascular risk, and osteoporosis. In these patients, fertility preservation is still associated with unfavorable pregnancy outcomes due to radiation-induced damage to the uterine and myometrial vessels, with increased risks of preterm labor and normal fetal growth alterations [44]. Infertility is frequently associated with high levels of stress, depression, and anxiety [22].

## 4. Conclusions

Although cervical cancer takes first place in terms of pathology in women, it is extremely rare in childhood and adolescence. Adenocarcinoma of the cervix and vagina is rare in pediatric populations, but a high incidence of stage III and stage IV disease can be observed. Non-HPV-related cervical cancer still has uncertain etiopathogenesis and nonspecific clinical manifestations that delay correct diagnosis. Very often, prolonged vaginal bleeding in teenagers is considered to be determined by the normal hormonal axis instability that can be observed during puberty. This aspect makes it challenging for clinicians attempting to determine a correct diagnosis in time, especially when the dimensions of the tumoral mass directly correlate itself with the survival rate and prognosis. Another important issue that needs to be considered in pediatric populations affected by cervical cancer is fertility preservation, which needs to be an integral part of a correct management strategy.

## Figures and Tables

**Figure 1 ijerph-19-16652-f001:**
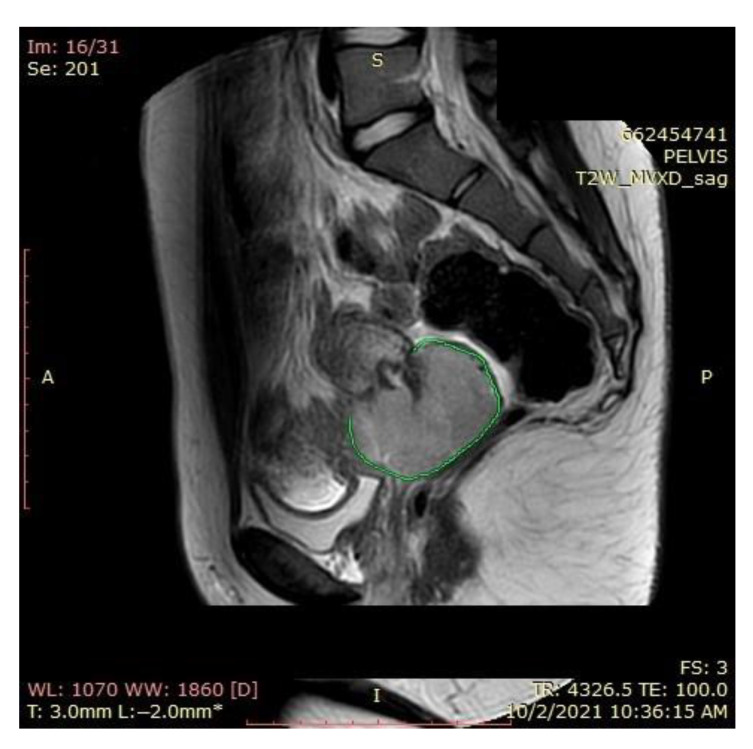
Pelvic sagittal T2–weighted sequence: the green highlighted area represents the circumferentially developed cervical tumor mass with intermediate T2 signal that includes both the endo- and the ectocervix protruding into the upper vagina.

**Figure 2 ijerph-19-16652-f002:**
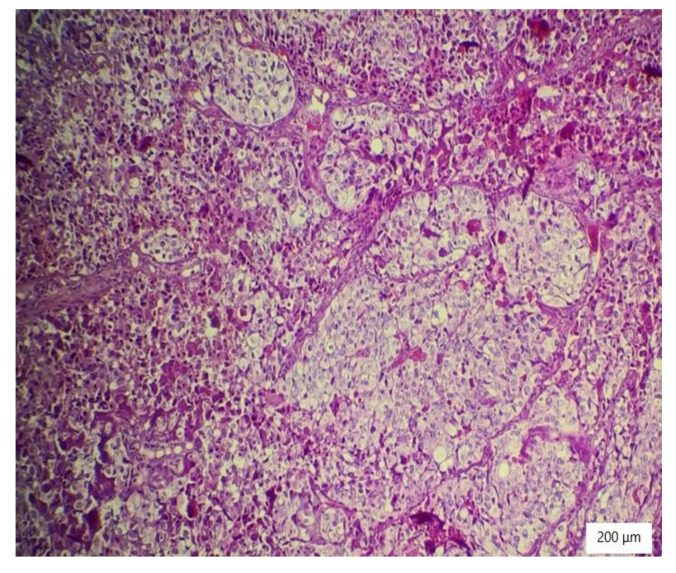
Solid architecture of the tumor with highly atypical cells with hyperchromatic nuclei and necrosis (HE ×100).

**Figure 3 ijerph-19-16652-f003:**
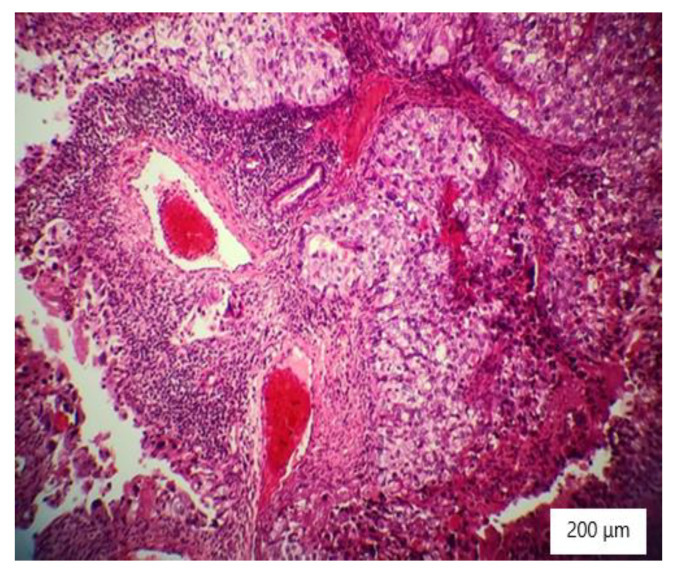
Tubulocystic pattern of tumor cells with clear cytoplasm, inflammatory infiltrate, necrosis, and hemorrhagic areas (HE ×100).

**Figure 4 ijerph-19-16652-f004:**
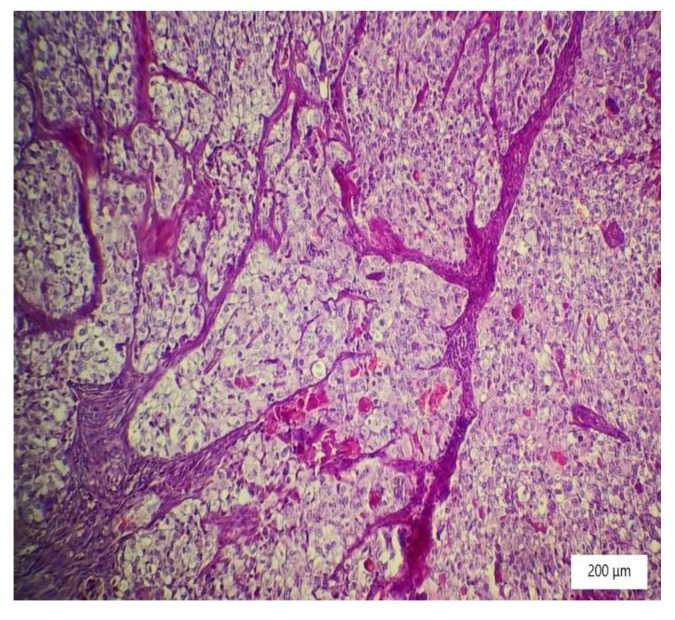
Tumor nests with large neoplastic cells with prominent cell borders, ovoid nuclei, clear cytoplasm, and stromal hyalinization (HE ×100).

**Figure 5 ijerph-19-16652-f005:**
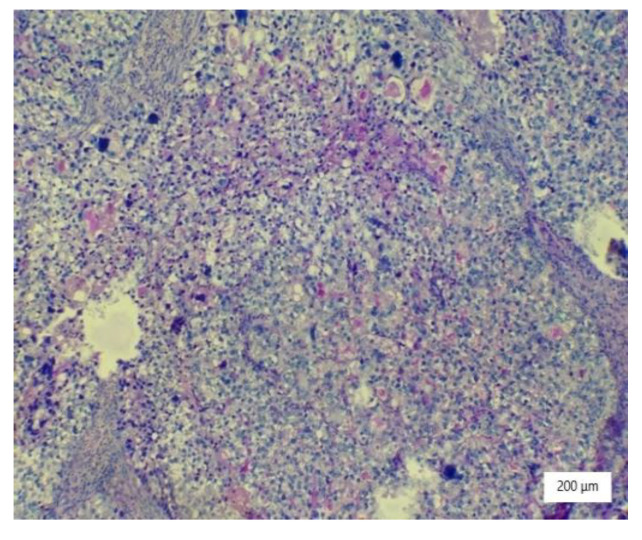
Positive periodic acid–Schiff staining in the cytoplasm (PAS ×100).

**Figure 6 ijerph-19-16652-f006:**
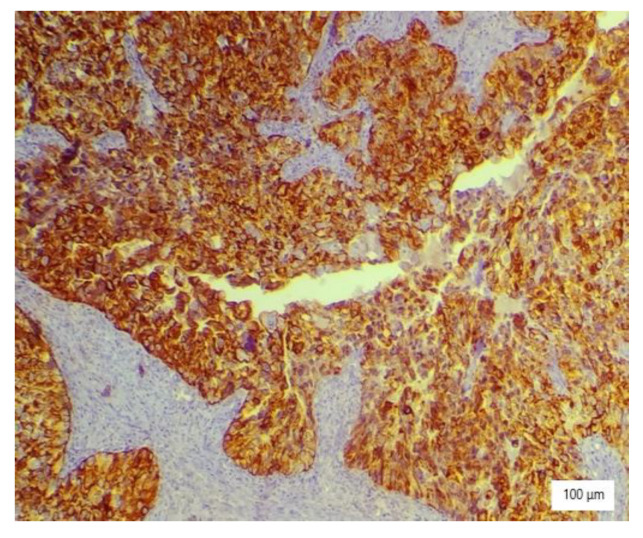
Diffuse AE1/AE3 immunostaining of large neoplastic cells with clear cytoplasm (×200).

**Figure 7 ijerph-19-16652-f007:**
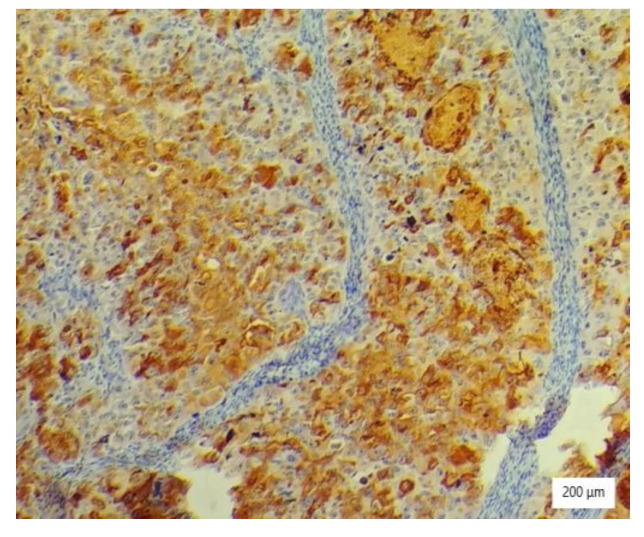
EMA immunostaining of tumor cells with high nuclear grade (×100).

**Figure 8 ijerph-19-16652-f008:**
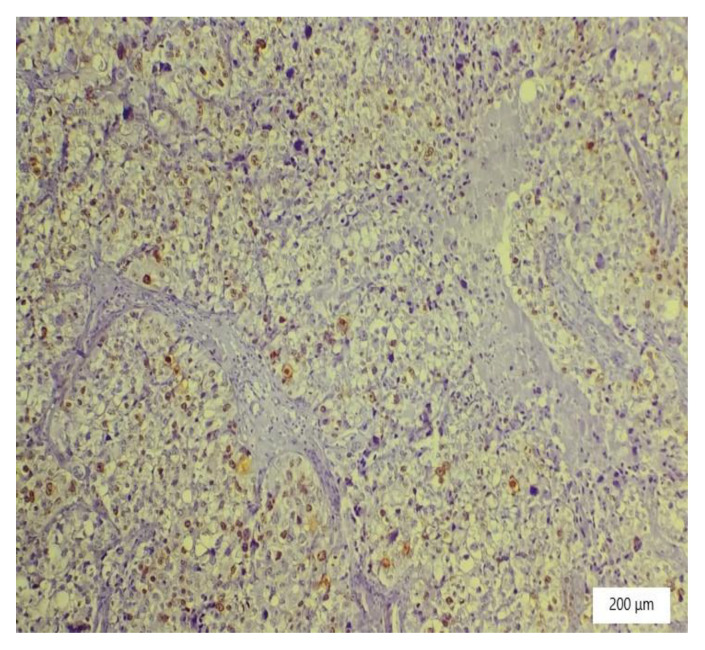
Ki 67 positivity in tumor cells (×100).

**Figure 9 ijerph-19-16652-f009:**
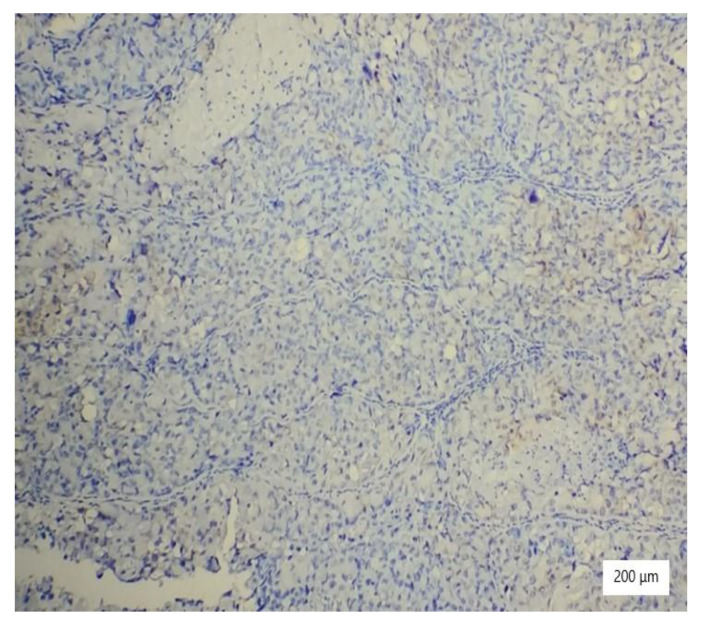
Variable, low expression pattern of P16 in tumor cells (×100).

**Figure 10 ijerph-19-16652-f010:**
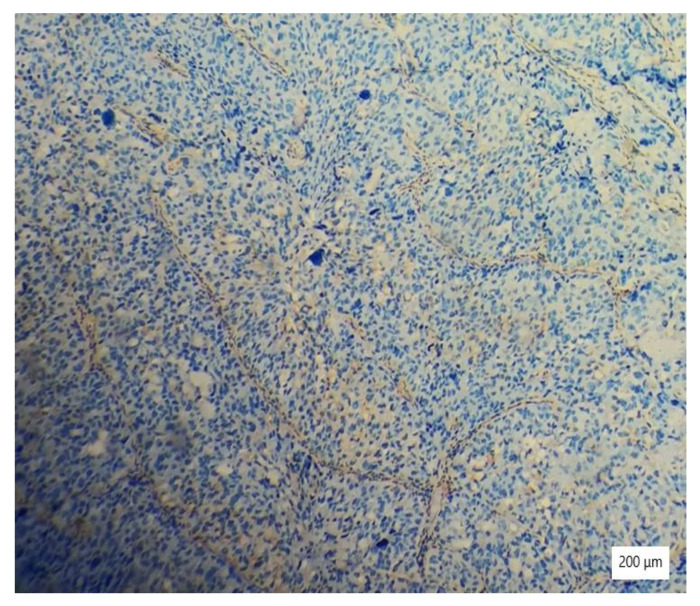
Negative vimentin in tumor cells (×100).

**Figure 11 ijerph-19-16652-f011:**
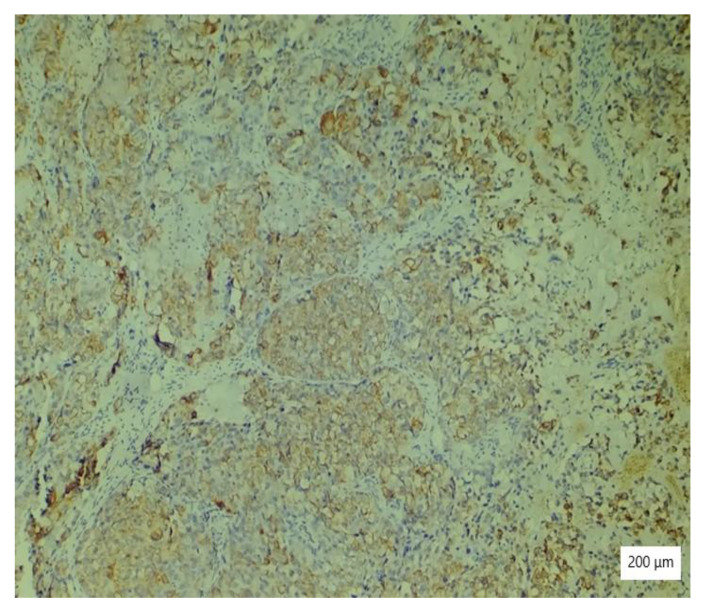
CK7 positivity in tumor cells (×100).

**Figure 12 ijerph-19-16652-f012:**
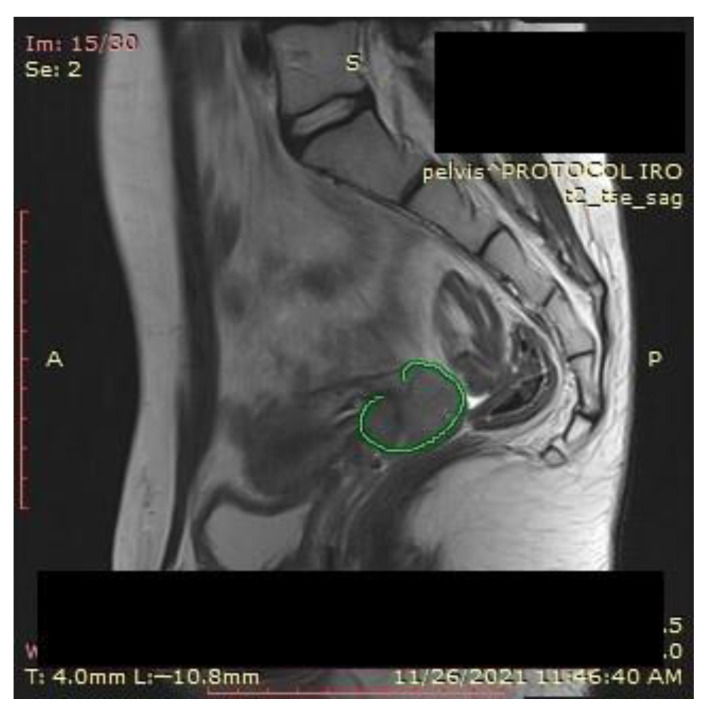
Favorable postradiotherapy response with the reduction in cervical mass (the green highlighted area represents the cervical tumoral mass).

**Figure 13 ijerph-19-16652-f013:**
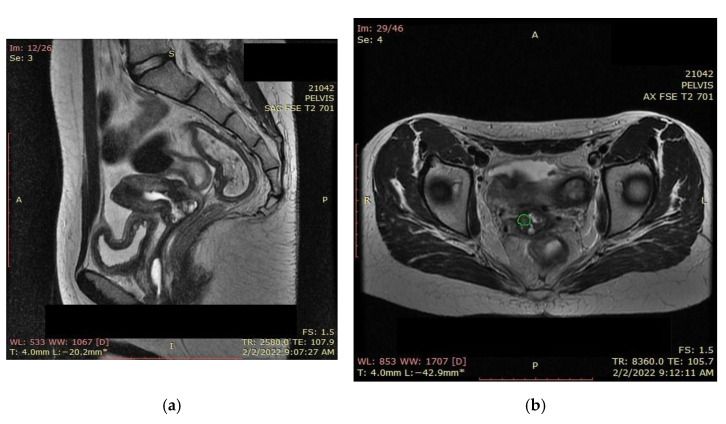
(**a**) MRI and (**b**) CT—significant favorable post-radiotherapy and chemotherapy response, with the persistence of a necrotic and ulcerative cervical area and a possible residual lesion on the right side of the cervix (the green highlighted area).

**Figure 14 ijerph-19-16652-f014:**
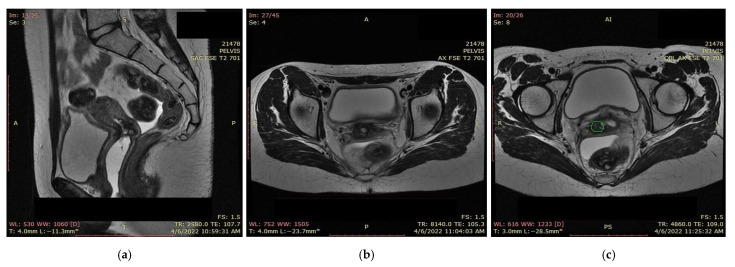
(**a**) MRI and (**b**,**c**) CT: almost completely undetectable lesion post-radiotherapy and chemotherapy response with the persistence of an involute area of necrosis and ulceration situated at on the topography of the former tumor (the green highlighted area).

**Table 1 ijerph-19-16652-t001:** Summary of previous reported cases of clear cell adenocarcinoma in teenagers during 1976–2022 (without DES exposure and non-HPV related).

Report (Author and Years of Publication)	No. of Cases	Age (Years)	Stage of FIGO
Noller et al. (1974) [12]	4	7 10 13 14	IA IIB IB IIA
Wesolowski et al. (1997) [13]	1	8	IB
Seki et al. (2003) [9]	1	18	IB2
Ding et al. (2004) [14]	1	19	IB2
Abu-Rustum et al. (2005) [15]	2	6 8	IB1 IB1
Ahrens et al. (2005) [16]	1	6	IB1
Chan et al. (2008) [17]	1	14	IIIA
Yabushita et al. (2008) [18]	1	17	IB1
Lester et al. (2010) [19]	1	6	IB1
Singh et al. (2011) [20]	1	13	IB1
Romero-Duran et al. (2012) [21]	1	11	unknown
Ansari et al. (2012) [22]	1	14	IIIA
Choi et al. (2013) [23]	1	15	IIA
Jiang et al. (2014) [24]	2	19 20	IIA2 IB1
Ronneberg et al. (2014) [25]	1	17	IB1
Baykara et al. (2014) [26]	2	14 16	IB2 IB1
Andi Asri et al. (2016) [27]	1	10	IB2
Singh et al. (2016) [28]	1	14	primary
Arora et al. (2017) [11]	1	1	I
Tantitamit et al. (2017) [29]	1	19	IB1
Su et al. (2020) [30]	1	6	IIA1
Levinson et al. (2020) [31]	1	15	relapsed
Zhang et al. (2021) [32]	1	19	IIIB
Liu et al. (2021) [33]	1	12	IIIC1

## Data Availability

The data used to support the findings of this study are available upon request from the authors.

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
