# Peer review of "Primary Clear Cell Adenocarcinoma of the Uterine Cervix in a 14-Year-Old Virgin Girl: Case Report"

_ijerph, 2022, doi:10.3390/ijerph192416652_

Round 1

Reviewer 1 Report

This is a case report based on a rare cervical cancer in a 14-year-old virgin girl. The author has explained the case in depth with all the results. This topic is relevant to the field as it demonstrates the diagnosis and treatment process of a rare carcinoma in a young patient.  It will be helpful in the treatment of these patients in the future.  And this is a rare case study and there are very few papers related to this topic. Their treatment process was successful in this case. All their conclusions were relevant to the results/evidence. We can clearly observe the reduction in the mass after surgical procedure. The references are appropriate. For tables and figures, Author can add vaginoscopy results as they reported clots and cauliflour like mass during test.

This case report is important as mentioned by the author there are very few similar cases in the literature. There are minor feedbacks as mentioned below:

1. Some of the sentences are in italics such as lines 45-48, 144-148, and 161-163. Check the whole manuscript thoroughly.

2. In line 208, It's bcl-2 overexpression instead of blc-2.  Also, referred articles do not discuss bcl-2 overexpression. Check the references again.

3. Author mentioned that they performed vaginoscopy which showed clots and a cauliflower-like mass. Can you add the data to the report?

4. In the manuscript, diethylstilbestrol is written in italics in a few places. 

5. non-HPV full form is not mentioned in the abstract.

Author Response

This case report is important as mentioned by the author there are very few similar cases in the literature. There are minor feedbacks as mentioned below:

  1. Some of the sentences are in italics such as lines 45-48, 144-148, and 161-163. Check the whole manuscript thoroughly.

We have corrected the sentences and the words in italics.

  1. In line 208, It's bcl-2 overexpression instead of blc-2.  Also, referred articles do not discuss bcl-2 overexpression. Check the references again.

We have corrected the mentioned mistake and checked the reference and modified it.

  1. Author mentioned that they performed vaginoscopy which showed clots and a cauliflower-like mass. Can you add the data to the report?

In the Department of Pediatric Surgery, Children’s Clinical Hospital “St. Maria” from Iasi where the vaginoscopy and biopsy were performed, there is no possibility to save photos or films of the laparoscopic or hysteroscopic interventions performed. That is the reason why we don’t have images from the vaginoscopy.

  1. In the manuscript, diethylstilbestrol is written in italics in a few places. 

We have corrected it.

  1. non-HPV full form is not mentioned in the abstract.

We mentioned it in the abstract.

Reviewer 2 Report

This is a case report, even though the pathology is not so common, the relevance and contribution to the area of knowledge is limited. The literature review could have been more extensive. More emphasis could have been placed on the fertility preservation aspect. Being only the presentation of one case, the value of its publication lies in the review of the associated literature. 

Author Response

This is a case report, even though the pathology is not so common, the relevance and contribution to the area of knowledge is limited. The literature review could have been more extensive. More emphasis could have been placed on the fertility preservation aspect. Being only the presentation of one case, the value of its publication lies in the review of the associated literature. 

As the second reviewer suggested, we have updated the literature, added a review in literature concerning DES unexposed non-HPV-related clear cell cervical carcinoma in teenagers and discussed fertility preservation aspects.

Reviewer 3 Report

This paper is a case report on a very extremely cervical clear cell adenocarcinoma. In particular, this is a young age patient with no history of DES exposure and successfully treated after radio-chemotherapy, which is thought to be worth sharing.

I would like to correct some of the points mentioned below.

1. It is better to use the abbreviation of the words mentioned earlier in the latter part.
p.1, line 33
: diethylstilbestrol (DES)
à DES

2. It is better to write the full name and the abbreviation together as the first word in the text.
p.2, line 48
:DES
à diethylstilbestrol (DES)

3. Check once again whether the invading part of the lesion is 2/3 of the upper vagina or 2/3 of the upper part of the vagina.
p.7 line 152

4. It would be good to mention the interval between radiation therapy and MRI. This is because there is a difference in the response rate between immediately after the end of treatment and after a certain period of time.
p.7, line 164

5. It would be nice to provide more specific information about brachytherapy. How much dose was given, and if possible, it would be good to add a brachytherapy dose distribution image.
p.8, line 173

6. If there is no current recurrence, it would be better to mention the follow-up period so that we can know the disease-free duration.
p.8, line 175

7. It is better to use the term uniformly.
p.9, line 245
: radiation-chemotherapy
à radio-chemotherapy

Author Response

I would like to correct some of the points mentioned below.

  1. It is better to use the abbreviation of the words mentioned earlier in the latter part.
    p.1, line 33: diethylstilbestrol (DES) à DES.

We have adjusted the mentioned mistake.

  1. It is better to write the full name and the abbreviation together as the first word in the text.
    p.2, line 48: DES à diethylstilbestrol (DES).

We have corrected the mentioned mistake.

  1. Check once again whether the invading part of the lesion is 2/3 of the upper vagina or 2/3 of the upper part of the vagina. p.7 line 152.

The tumor occupied the upper 1/3 of the vagina. We rephrased the sentence to make it clear.

  1. It would be good to mention the interval between radiation therapy and MRI. This is because there is a difference in the response rate between immediately after the end of treatment and after a certain period of time. p.7, line 164.

The MRI was performed immediately the next day after finishing the radiotherapy and chemotherapy sessions.

  1. It would be nice to provide more specific information about brachytherapy. How much dose was given, and if possible, it would be good to add a brachytherapy dose distribution image.
    p.8, line 173.

We have mentioned in the article more detailed information about the brachytherapy. She underwent three sessions of 3D intracavitary and interstitial cervico-uterine brachytherapy after the insertion of 8 needles Ring and Tandem applicator. All three sessions took place 2 days apart in December 2021 the first one with 7.5 Gy and the next two with 8 Gy. The brachytherapy sessions were done in a private clinic in another city and we don’t have access to dose distribution images.

  1. If there is no current recurrence, it would be better to mention the follow-up period so that we can know the disease-free duration. p.8, line 175

We have mentioned the follow-up period. Until the submission of the article no recurrence was detected using periodical MRI and CT evaluation of the patient (from April until September 2022).

  1. It is better to use the term uniformly.
    p.9, line 245: radiation-chemotherapy à radio-chemotherapy.

We have corrected the above mentioned terms.